# Evaluation of Various Alternative Economical and High Throughput SARS-CoV-2 Testing Methods within Resource-Limited Settings

**DOI:** 10.3390/ijms232214350

**Published:** 2022-11-18

**Authors:** Zamathombeni Duma, Veron Ramsuran, Anil A. Chuturgoon, Vinodh A. Edward, Pragalathan Naidoo, Zilungile L. Mkhize-Kwitshana

**Affiliations:** 1Disciplines of Medical Microbiology, Nelson R. Mandela School of Medicine, University of KwaZulu-Natal, Umbilo, Durban 4041, South Africa; 2Division of Research Capacity Development, South African Medical Research Council (SAMRC), Tygerberg, Cape Town 7505, South Africa; 3Disciplines of Medical Microbiology, Howard College, University of KwaZulu-Natal, Glenwood, Durban 4041, South Africa; 4Disciplines of Medical Biochemistry, Howard College, University of KwaZulu-Natal, Glenwood, Durban 4041, South Africa; 5The Aurum Institute, 29 Queens Road, Parktown, Johannesburg 2193, South Africa; 6School of Health Sciences, College of Health Sciences, University of KwaZulu-Natal, Westville, Durban 3629, South Africa; 7Department of Environmental Health Sciences, Yale School of Public Health, New Haven, CT 06511, USA

**Keywords:** SARS-CoV-2, diagnostic testing, low-middle income countries, resource-limited settings, alternative cost-effective and high throughput testing approaches

## Abstract

The severe acute respiratory syndrome coronavirus 2 (SARS-CoV-2) outbreak posed a challenge for diagnostic laboratories worldwide, with low-middle income countries (LMICs) being the most affected. The polymerase chain reaction (PCR) is the gold standard method for detecting SARS-CoV-2 infection. However, the challenge with this method is that it is expensive, which has resulted in under-testing for SARS-CoV-2 infection in many LMICs. Hence, this study aimed to compare and evaluate alternative methods for the mass testing of SARS-CoV-2 infection in laboratories with limited resources to identify cost-effective, faster, and accurate alternatives to the internationally approved kits. A total of 50 residual nasopharyngeal swab samples were used for evaluation and comparison between internationally approved kits (Thermo Fisher PureLink™ RNA Isolation Kit and Thermo Fisher TaqPath™ COVID-19 Assay Kit) and alternative methods (three RNA extraction and four commercial SARS-CoV-2 RT-PCR assay kits) in terms of the cost analysis, diagnostic accuracy, and turnaround time. In terms of performance, all of the alternative RNA extraction methods evaluated were comparable to the internationally approved kits but were more cost-effective (Lucigen QuickExtract™ RNA Extraction Kit, Bosphore EX-Tract Dry Swab RNA Solution and Sonicator method) and four commercial SARS-CoV-2 RT-PCR assay kits (Nucleic Acid COVID-19 Test Kit (SARS-CoV-2), abTES^TM^ COVID-19 qPCR I Kit, PCL COVID19 Speedy RT-PCR Kit, and PCLMD nCoV One-Step RT-PCR Kit) with a sensitivity range of 76–100% and specificity of 96–100%. The cost per sample was reduced by more than 50% when compared to internationally approved kits. When compared to the Thermo Fisher PureLink™ Kit and Thermo Fisher TaqPath™ COVID-19 Assay Kit, the alternative methods had a faster turnaround time, indicating that laboratories with limited resources may be able to process more samples in a day. The above-mentioned cost-effective, fast, and accurate evaluated alternative methods can be used in routine diagnostic laboratories with limited resources for mass testing for SARS-CoV-2 because these were comparable to the internationally approved kits, Thermo Fisher PureLink™ Kit and Thermo Fisher TaqPath™ COVID-19 Assay Kit. The implementation of alternative methods will be the most cost-effective option for testing SARS-CoV-2 infection in LMICs.

## 1. Introduction

The severe acute respiratory syndrome coronavirus 2 (SARS-CoV-2), the virus that causes coronavirus disease 2019 (COVID-19) [1], has spread globally since its first recorded outbreak in Wuhan, China in December 2019 [2,3]. As of 8 August 2022, there have been over 589 million confirmed cases and 5.4 million deaths worldwide [4].

Diagnostic testing remains critical in controlling the SARS-CoV-2 outbreaks, allowing patients to be cared for while also simultaneously providing decision-makers with critical information for test-trace isolation programs [5,6]. Most countries have experienced increased demand for SARS-CoV-2 diagnostic testing, with some countries unable to meet the demand. This is one of the major challenges, especially in low-middle income countries (LMICs) where unstable health systems and reliance on global supply chains have frequently prevented people from accessing critical tests for detecting SARS-CoV-2 infection [7,8]. In many LMICs, insufficient testing may have resulted in an underestimation of SARS-CoV-2 infections. What is concerning is that the pandemic’s long-term impact on individuals and communities in LMICs remains uncertain as the number of confirmed cases continues to increase [9].

SARS-CoV-2 is an enveloped virus with a single positive-sense RNA genome [1]. Furthermore, the SARS-CoV-2 genome contains non-structural open reading frames (ORF1ab), which are polypeptide coding genes that are translated from genomic RNA [10,11,12]. It also contains four structural proteins, namely, the spike (S), membrane (M), envelope (E), and nucleocapsid (N) proteins that contribute to the SARS-CoV-2 overall structure [13,14]. SARS-CoV-2 infection can be detected using two different types of tests: real-time reverse transcription polymerase chain reaction (RT-PCR) and antigen rapid tests [15,16]. Antigen rapid tests are less expensive and provide results faster than RT-PCR. However, antigen rapid tests, on the other hand, are less accurate in detecting SARS-CoV-2 infection, especially in asymptomatic individuals and those with low SARS-CoV-2 viral load [17,18,19]. According to a Cochrane systematic review of 22 antigen rapid test trials for detecting SARS-CoV-2 infection, the antigen rapid test showed an average sensitivity of 56.2% [20]. Furthermore, antigen rapid testing was found to have a higher risk of false negatives than molecular RT-PCR tests, with some evidence indicating false negative rates as high as 50%, therefore, a confirmatory RT-PCR test is still recommended [20,21].

The RT-PCR assay is the most accurate test for detecting SARS-CoV-2, hence it is regarded as a gold standard diagnostic procedure for diagnosing SARS-CoV-2 infection [22,23]. The *N* gene, *S* gene, *E* gene, and *ORF1ab* gene are the most tested target genes for SARS-CoV-2 infection using the RT-PCR assay [24,25]. The RT-PCR assay has superior sensitivity and specificity in comparison to antigen and antibody rapid tests, however, it is highly specialized and expensive, especially for LMICs. This is partly due to a lack of local capacity in these countries to produce their analytical instrument and reagents for RT-PCR-based SARS-CoV-2 testing [8,23]. As a result, mass testing for SARS-CoV-2 infection in LMICs poses a challenge. Of note, high-income countries (HICs) test more samples daily to control the spread of the SARS-CoV-2 virus, whereas LMICs test fewer samples due to financial constraints [26]. It is therefore essential to evaluate the alternative methods for SARS-CoV-2 testing that are simple, cost-effective, and produce high throughput results in a short period of time in a laboratory within a resource-limited setting. 

This study aimed to compare and evaluate whether alternative RNA extraction methods (Lucigen QuickExtract™ RNA Extraction Kit, Bosphore EX-Tract Dry Swab RNA Solution, and Sonicator method) are cost-effective, faster, and have clinical accuracy comparable to an internationally approved kit (Thermo Fisher PureLink™ Kit, Pleasanton, CA, USA) and to also assess whether these extraction methods can be used interchangeably. Furthermore, the performance of commercially available RT-PCR SARS-CoV-2 assay kits (Nucleic Acid COVID-19 Test Kit (SARS-CoV-2), abTESTM COVID-19 qPCR I Kit, PCL COVID19 Speedy RT-PCR Kit, and PCLMD nCoV One-Step RT-PCR Kit) was compared to the internationally approved kit (Thermo Fisher TaqPath™ COVID-19 Assay Kit) in terms of the sensitivity, specificity, costs, and turnaround times.

## 2. Results

In the first part of this study, the Thermo Fisher PureLink™ Kit was compared to three alternative RNA extraction methods. In the second part of this study, the Thermo Fisher TaqPath™ COVID-19 Assay Kit was compared to four alternative commercially available RT-PCR SARS-CoV-2 assay kits. The overall aim of this study was to assess and compare the clinical performance of these methods as well as determine whether they could be used interchangeably to help increase the testing capacity for SARS-CoV-2 infection in LMICs.

### 2.1. Comparison of RNA Extraction Methods

The multiplex Thermo Fisher TaqPath™ COVID-19 Assay Kit was used to assess the RNA extraction efficiency of each extraction method. For the positive (infected) group, the efficiency of the RNA extraction methods was assessed by comparing the mean Ct values of the SARS-CoV-2 targeting genes (*N*, *S*, *ORF1ab*), and the internal control (*MS2*) between the alternative RNA extraction methods and the Thermo Fisher PureLink™ Kit. The efficiency of the RNA extraction methods was assessed for the control (uninfected) group by comparing the mean Ct values of the internal control (*MS2*) between the alternative RNA extraction methods and the Thermo Fisher PureLink™ Kit. The PCR was run in triplication for each sample for the purpose of method comparison, and the average Ct value result was used for each sample. The results were considered positive if the cycle threshold value (Ct value) for all three target genes for SARS-CoV-2 and the internal control were less than 40 (Ct ≤ 40). When all of the SARS-CoV-2 target genes were negative and the internal control was positive (Ct ≤ 40), the results were considered negative. When the internal control was negative, the results were considered invalid due to the inefficacy of the RNA extraction method. 

The current results showed that all the RNA extraction methods could be used to extract high-quality RNA for the testing of SARS-CoV-2 infection because the Ct values of SARS-CoV-2 targeting genes (*N*, *S*, *ORF1ab*), and the internal control (*MS2*) for all RNA extraction methods were less than 35 (Ct ≤ 35) (Table 1). There was also no significant difference in the mean Ct values (*p* ≤ 0.05 **) between the alternative RNA extraction methods (Lucigen QuickExtract™ RNA Extraction Kit (Parmenter St Middleton, WI, USA), Bosphore EX-Tract Dry Swab RNA (New Ash Green Longfield, England) and Sonicator method (Europe) and the internationally approved kit (Thermo Fisher PureLink™ Kit) for the SARS-CoV-2 targeting genes (*N*, *S*, *ORF1ab*) and internal control (*MS2*) (Table 1 and Figure 1A–D). However, the results of one of the alternate RNA extraction methods, the Sonicator method, showed that there was a statistically significant difference (*p* ≤ 0.05) in Ct values between the Sonicator method and the Thermo Fisher PureLink™ Kit for the mean Ct value of the *ORF1ab* gene (Table 1 and Figure 1C). 

### 2.2. Comparison of Sensitivity and Specificity for RNA Extraction Methods

Based on the small sample size, caution should be taken when using this study’s sensitivity and specificity results for the method comparison. The accuracy of each RNA extraction method used in testing SARS-CoV-2 infection was assessed by comparing the calculated sensitivity and specificity of alternative RNA extraction methods to the internationally approved kit. In a total of 25 nasopharyngeal swab samples for the positive (infected) group, the results of 25 positive samples extracted using two alternative extraction methods (Bosphore EX-Tract Dry Swab RNA Solution and Lucigen QuickExtract™ RNA Extraction Kit) matched the results of 25 positive samples extracted using the Thermo Fisher PureLink™ Kit. However, only 24 of the 25 positive nasopharyngeal swab sample results extracted using the Sonicator method (alternative RNA extraction method) matched the 25 positive sample results extracted using the Thermo Fisher PureLink™ Kit, with one sample result being invalid.

Additionally, in a total of 25 nasopharyngeal swab samples for a control group, all 25 negative samples extracted using alternative RNA extraction methods matched the 25 negative sample results that were extracted using the Thermo Fisher PureLink™ Kit. The sensitivity ranged from 96 to 100% for all alternative RNA extraction methods, while the specificity was 100% (Table 2). 

### 2.3. Overview and Comparison of the Commercially Available SARS-CoV-2 RT-PCR Assay Kits

According to the manufacturer’s guidelines for each kit, the test results were considered positive when all of the SARS-CoV-2 target genes and internal control used in the particular kit were detected at the same time (Ct ≤ 40 or Ct < 35). When all of the SARS-CoV-2 target genes were negative (Ct > 40) and the internal control was positive (Ct ≤ 40 or Ct < 35), the results were considered negative. None of the manufacturers were involved in the analysis and interpretation of the results. Table 3 summarizes the requirements for all of the commercially available RT-PCR SARS-CoV-2 test kits used to detect the SARS-CoV-2 target genes, as specified in the documentation for each RT-PCR SARS-CoV-2 assay kit.

This study compared the calculated sensitivity and specificity of the alternative commercial RT-PCR assay kits (Nucleic Acid COVID-19 Test Kit (SARS-CoV-2), abTES^TM^ COVID-19 qPCR I Kit, PCL COVID19 Speedy RT-PCR Kit, and PCLMD nCoV One-Step RT-PCR Kit) to the internationally approved kit (Thermo Fisher TaqPath™ COVID-19 Assay Kit). The sensitivity and specificity of the evaluated commercially available RT-PCR SARS-CoV-2 assay kits varied, with the PCLMD nCoV One-Step RT-PCR Kit having the highest sensitivity and specificity (96% and 100%, respectively) and the abTES^TM^ COVID-19 qPCR I Kit having the lowest sensitivity and specificity (76% and 96%, respectively). The results are presented in Table 4.

### 2.4. Cost Analysis, Simplicity and the Turnaround Time for Each Evaluated Method and Internationally Approved Kits

The costs of the RNA extraction methods and commercially available RT-PCR SARS-CoV-2 assay kits were calculated and compared using the pricing of the reagents, consumables, and equipment. The processing time for each RNA extraction method and the observed run time for each commercial RT-PCR test kit were calculated to determine the turnaround time for each RNA extraction method and the commercially available RT-PCR SARS-CoV-2 assay kits. The prices for the RNA extraction methods and commercially available RT-PCR SARS-CoV-2 assay kits are specific to South Africa.

The results in Table 5 show that the RNA extraction methods varied in terms of cost, processing time, and procedure simplicity. The Thermo Fisher PureLink™ Kit was the most expensive and had a per sample cost of USD ~2.96. The Sonicator method (an alternative RNA extraction method) was the least expensive, with a cost per sample of USD ~0.18. The processing time results for each RNA extraction method were as follows: Thermo Fisher PureLink™ Kit (~1 h); Sonicator method (~30 min); Bosphore EX-Tract Dry Swab RNA Solution (~15 min); Lucigen QuickExtract™ RNA Extraction Kit (~15 min). The Bosphore EX-Tract Dry Swab RNA Solution and Lucigen QuickExtract™ RNA Extraction Kit were considered to be the two simplest methods (three steps each) for extracting RNA nucleic acid used for testing SARS-CoV-2 infection.

The results of the cost analysis for each commercially available RT-PCR SARS-CoV-2 assay kit showed that all alternative commercial RT-PCR SARS-CoV-2 assay kits were the least expensive when compared to the Thermo Fisher TaqPath™ COVID-19 Assay Kit. The results of the cost of the commercially available RT-PCR SARS-CoV-2 assay kits are presented in Table 6. Additionally, the results indicated that there was a slight difference in the observed run time between the Thermo Fisher TaqPath™ COVID-19 Assay Kit and the three commercial SARS-CoV-2 RT-PCR assay kits (Nucleic Acid COVID-19 Test Kit (SARS-CoV-2), abTES^TM^ COVID-19 qPCR I Kit, PCL COVID19 Speedy RT-PCR Kit). The observed run times between the Thermo Fisher TaqPath™ COVID-19 Assay Kit (64 min) and the PCLMD nCoV One-Step RT-PCR Kit (137 min), one of the alternative commercial SARS-CoV-2 RT-PCR assay kits, were, however, significantly different (Table 6).

## 3. Discussion

As countries around the world continue to search for effective treatment and eradication of the SARS-CoV-2 virus, diagnostic testing is still one of the most effective ways to track the spread of the virus and subsequently implement appropriate preventative measures [28,29]. Many factors, particularly financial and infrastructural resources, limit the quantum of testing in LMICs. The main goal of this study was to evaluate the cost-effective, accurate, and faster alternative methods that could assist in increasing the testing capacity for SARS-CoV-2 infection in laboratories within limited-resource settings. Results showed that the alternative RNA extraction methods (Sonicator method, Lucigen QuickExtract™ RNA Extraction Kit, and Bosphore EX-Tract Dry Swab RNA Solution) were qualitatively comparable to the internationally approved kit Thermo Fisher PureLink™ Kit. Likewise, the performance characteristics of the alternative commercial RT-PCR SARS-CoV-2 assay kits (Nucleic Acid COVID-19 Test Kit (SARS-CoV-2), abTES^TM^ COVID-19 qPCR I Kit, PCL COVID19 Speedy RT-PCR Kit, and PCLMD nCoV One-Step RT-PCR Kit) were as follows: (i) sensitivity (76–96%); (ii) specificity (96–100%); (iii) negative predictive value (4–24%); and (iv) positive predictive value (4%). 

The efficiency of each RNA extraction method was assessed by comparing the performance of alternative RNA extraction methods to the Thermo Fisher PureLink™ Kit (internationally approved kit). In the present study results, there was no statistically significant difference in the mean Ct value between the Thermo Fisher PureLink™ Kit and the alternative extraction methods (Ct value ≤ 35. All of the alternative RNA extraction methods had high specificity (100%) and sensitivity (96–100%). The statistically significant difference in the Ct value for the *ORF1ab* gene between the Sonicator extraction method and the Thermo Fisher PureLink™ Kit had no appreciable effect on the Sonicator extraction method’s high sensitivity. The results showed that the efficiency and recovery rates of the alternative RNA extraction methods were satisfactory to be used for RNA extraction in testing for the SARS-CoV-2 infection.

When the cost-effectiveness, processing time, and simplicity of each RNA extraction method were compared, it was found that both the Bosphore EX-Tract Dry Swab RNA Solution and Lucigen QuickExtract™ RNA Extraction Kit were simpler, faster, accurate, and cheaper, despite the fact that neither of these extraction methods had any inhibitor removal. These two alternative methods had three procedural steps and took approximately 15 min to extract RNA from 25 nasopharyngeal samples compared to the Thermo Fisher PureLink™ Kit, which had five procedural steps and took approximately an hour to process 25 nasopharyngeal samples. The Thermo Fisher PureLink™ Kit was the most expensive (Table 5). Despite being more time-consuming, the Sonicator method was the cheapest of the two alternative RNA extraction methods (Bosphore EX-Tract Dry Swab RNA Solution and Lucigen QuickExtract™ RNA Extraction Kit) and the internationally approved kit (Thermo Fisher PureLink™ Kit). When comparing the cost of the Sonicator method (USD ~0.18 per sample) to the Thermo Fisher PureLink™ Kit (USD ~2.96 per sample), there was more than a 94% price reduction with Sonicator methods, making the Sonicator method the cheapest method. Hence, the Sonicator extraction method may be a good choice, especially considering that laboratories in low-income countries (LICs) have limited resources for the mass testing of SARS-CoV-2 infection, and setting up a laboratory for testing SARS-CoV-2 infection in these countries is a challenge. This study recommends that LICs use the Sonicator method. Consequently, SARS-CoV-2 mass testing will be improved, and viral transmission will be optimally monitored for control and reduction.

Viral RNA extraction is necessary for the RT-PCR tests to be performed [30]. The Lucigen QuickExtract™ RNA Extraction Kit was the method of choice for RNA extraction in this study. The extracted RNA was required for the evaluation and comparison of commercial RT-PCR SARS-CoV-2 kits. The Lucigen QuickExtract™ RNA Extraction Kit was chosen due to its simplicity, speed, and low cost. When the clinical accuracy of the alternative commercial RT-PCR SARS-CoV-2 assay kits was compared to the Thermo Fisher TaqPath™ COVID-19 Assay Kit, the study found that all four alternative commercially available RT-PCR SARS-CoV-2 assay kits had good diagnostic sensitivity ranging from 76–96% and a specificity of 96–100% (Table 5), with the PCLMD nCoV One-Step RT-PCR Kit having superior sensitivity compared to other the three alternative commercially available RT-PCR SARS-CoV-2 assay kits. The high specificity of 100% reported on the manufacturer’s package insert for all RT-PCR SARS-CoV-2 assay kits (Nucleic Acid COVID-19 Test Kit (SARS-CoV-2), abTES^TM^ COVID-19 qPCR I Kit, PCL COVID19 Speedy RT-PCR Kit, PCLMD nCoV One-Step RT-PCR Kit, and an internationally approved kit (Thermo Fisher TaqPath™ COVID-19 Assay Kit) matched the present study’s high specificity results (96–100%). All of the evaluated alternative RT-PCR SARS-CoV-2 assay kits and the internationally approved kit had a sensitivity of 92.89–100%, as specified on the manufacturer’s package insert, which matched the present study’s high sensitivity results (76–96%).

Interestingly, the inclusion of the *N* target gene on these purchased three alternative commercial RT-PCR SARS-CoV-2 assay kits as well as the Thermo Fisher TaqPath™ COVID-19 Assay Kit could be one of the main contributions that helped improve the sensitivity of these assay kits. According to published studies, the *N* gene is the most sensitive target gene for detecting SARS-CoV-2 because it contains a greater number of subgenomic *N* gene messenger RNAs compared to other target genes. [31,32]. Furthermore, the *ORF1ab* gene has been identified as the target gene with the highest contribution to specificity in the RT-PCR assay kits for the detection of SARS-CoV-2 infection. This is because the *ORF1ab* gene is the most conserved compared to other target genes such as the *N* or *E* genes [33,34]. The results demonstrated a good match between all four alternative commercial RT-PCR SARS-CoV-2 assays kits and the internationally approved Thermo Fisher TaqPath™ COVID-19 Assay Kit (100% sensitivity and 100% specificity) and can therefore be recommended for use interchangeably in routine diagnostic laboratories.

When comparing the costs of the Thermo Fisher TaqPath™ COVID-19 Assay Kit and alternative commercial RT-PCR SARS-CoV-2 assay kits, the Thermo Fisher TaqPath™ COVID-19 Assay Kit (USD ~14.80 per sample) was the most expensive of all of the commercial RT-PCR SARS-CoV-2 assay kits, with alternative commercial RT-PCR SARS-CoV-2 assay kit prices ranging between a USD ~4.44 and USD ~9.83 cost per sample. In terms of the observed run time, there seemed to be little difference between the Thermo Fisher TaqPath™ COVID-19 Assay Kit and the other three commercial RT-PCR SARS-CoV-2 kits (Nucleic Acid COVID-19 Test Kit (SARS-CoV-2), PCL COVID19 Speedy RT-PCR Kit, and PCLMD nCoV One-Step RT-PCR Kit), and the turn-around time was shorter for all these methods. Despite having the longest run time (137 min) of all the evaluated commercial RT-PCR SARS-CoV-2 assay kits, the PCMLD nCov One-Step RT-PCR Kit had some advantages, being cheaper and having a higher diagnostic sensitivity and specificity. For these reasons, the use of a Thermo Fisher TaqPath™ COVID-19 Assay Kit or low-cost alternative commercial RT-PCR SARS-CoV-2 assay kits may help to increase the mass testing for SARS-CoV-2 infection within a limited resource laboratory setting. 

The sample size was one of the study’s limitations due to the challenge of a limited cost budget. Therefore, a larger sample size is required for future studies in order to validate the current study results. Nonetheless, we believe that useful information has been established. Furthermore, this work is not only relevant for SARS-CoV-2 testing in LMICs, but also to the fact that scientists predict more pandemics as there is increased interaction between the environment, wildlife, and humans [35,36]. This then warrants that LMICs prepare for such, and one of the requisites for preparedness is cost-effective laboratory testing capabilities.

## 4. Material and Methods

This was a retrospective study to compare various methods to identify cost-effective methods that can be used by low-middle income countries. This study used residual nasopharyngeal swab samples from adult (over the age of 18) male and female participants who had SARS-CoV-2 symptoms or not. The study was approved by the University of KwaZulu Natal Biomedical Research Ethics Committee (BREC/00003671/2021) with permission to use residual nasopharyngeal samples with blinding to protect patient identity from both the BREC and the Global Health Innovation (GHI) laboratory, a subsidiary of the Aurum Institute, Johannesburg, South Africa.

### 4.1. Clinical Specimens

A total of 50 residual nasopharyngeal swab samples were used, which were initially collected from South African patients to test for SARS-CoV-2 infection by trained health care workers. Dry sterile nasopharyngeal swabs were used to test for SARS-CoV-2 infection, and the samples were transported in a cooler bag with ice to the GHI laboratory. Furthermore, for the comparison of methods in this study, the participants’ residual nasopharyngeal swab samples that were in deionized water and first tested for SARS-CoV-2 infection by the GHI laboratory were used. The GHI laboratory is accredited by the South African National Accreditation System (SANAS) for diagnostic testing. The 50 residual samples were subdivided into two groups: (i) Group 1: 25 nasopharyngeal residual samples from the SARS-CoV-2 infected patients (positive group), and (ii) Group 2: 25 nasopharyngeal residual samples from the SARS-CoV-2 uninfected patients (control group). All specimens were processed in a biosafety cabinet level 3 (BSL 3) facility with full personal protective equipment (PPE). 

This study compared the RNA extraction methods and commercially available SARS-CoV-2 RT-PCR assay kits using the same residual nasopharyngeal swab samples. The standard operating procedure in this study was normalized by using the same residual nasopharyngeal swab samples stored under the same conditions. In addition, the residual nasopharyngeal swabs used for method comparison were chosen from participants with almost identical Ct values for the SARS-CoV-2 target genes (*N*, *S*, *ORF1ab*) and internal control (MS2) as well as the high viral load for the positive group. Participants with almost identical Ct values for MS2 (internal control) were also chosen for the negative group. The residual nasopharyngeal swab samples were stored in a locked −80 °C bio-freezer to ensure the stability and integrity of the samples. 

### 4.2. Comparison of SARS-CoV-2 RNA Extraction Methods

The Thermo Fisher PureLink^TM^ Kit was chosen in this study as the gold standard method for the comparison of RNA extraction methods (Lucigen QuickExtract™ RNA Extraction Kit, Bosphore EX-Tract Dry Swab RNA Solution, and Sonicator method) because of its superior clinical accuracy and it is an internationally approved kit. Furthermore, the Thermo Fisher TaqPath^TM^ COVID-19 Assay Kit was also chosen as the gold standard assay kit for assessing the efficacy of each RNA extraction method and as a comparison method for assessing the clinical performance of alternative commercially available SARS-CoV-2 RT- PCR assay kits (Nucleic Acid COVID-19 Test Kit (SARS-CoV-2), abTES^TM^ COVID-19 qPCR I Kit, PCL COVID19 Speedy RT-PCR Kit, and PCLMD nCoV One-Step RT-PCR Kit) because of the following advantages: (i) The TaqPath™ COVID-19 Assay Kit can detect SARS-CoV-2 infection by identifying the presence of three gene targets from the virus’s *S*, *N,* and *ORF1ab* regions [37]; (ii) even if one of the targets is altered by a mutation, the test can provide reliable results, and (iii) the World Health Organization (WHO), Food and Drug Administration (FDA, Centers for Disease Control and Prevention (CDC), and European Centers for Disease Control (ECDC) have acknowledged the Thermo Fisher TaqPath Assay for using the S-gene target failure (SGTF) of PCR assays as a proxy for the variation aided in the diagnosis of Omicron [38,39]. Therefore, it is an internationally approved kit.

The RNA extraction methods used for comparison purposes include the Thermo Fisher PureLink™ Kit, an internationally approved kit, and the alternative extraction methods (Lucigen QuickExtract™ RNA Extraction Kit, Bosphore EX-Tract Dry Swab RNA Solution, and Sonicator method). Initially, 5 µL MS2 was added as an internal control to all of the Eppendorf microtubes (Merck, Darmstadt, Germany) containing nasopharyngeal swab samples in 300 µL deionized water. Then, the mixture was vortexed (Scientific Industries Inc., Bohemia, NY, USA) for 2 min to homogenize the samples. Thereafter, the homogenized samples were further used for the comparison of the methods (RNA extraction methods and commercially available SARS-CoV-2 RT-PCR assay kits).

#### 4.2.1. Thermo Fisher PureLink™ Kit

RNA was extracted according to the manufacturer’s instructions using the Thermo Fisher PureLink™ Kit (Thermo Fischer Scientific, Pleasanton, CA, USA, Cat No. A47813 and A47814). A total of 200 µL of the homogenized samples were used for manual RNA extraction. The procedures for RNA extraction with this kit included proteinase K buffer digestion, the addition of the binding bead solution, washing of the beads three times, and elution of the nucleic acid (RNA). The working sample volume of the extracted and eluted RNA was 50 µL. The Applied Biosystems Real-Time thermal cycler (RT-PCR) instrument (Thermo Fisher Scientific, Waltham, MA, USA) was used for the amplification and detection of SARS-CoV-2 target genes (*S*, *N,* and *ORF1ab*).

#### 4.2.2. Lucigen QuickExtract™ RNA Extraction Kit

Following the manufacturer’s instructions, RNA was extracted using the Lucigen, QuickExtract™ RNA Extraction Kit (LGC Biosearch Technologies, Parmenter St Middleton, WI, USA, Cat No. QER090150) with the minor protocol modification described below: 20 µL of the homogenized nasopharyngeal swab samples was added to MicroAmp 8-tube strips with 20 µL of the Lucigen, QuickExtract™ RNA Extraction solution to extract RNA. To inactivate the virus, the extracted sample was placed on the Applied Biosystems heat cycler for 5 min at 95 °C. The Applied Biosystems real-time thermal cycler (RT-PCR) instrument (Thermo Fisher Scientific, Waltham, MA USA) was used for the amplification and detection of SARS-CoV-2 target genes from the extracted RNA samples. 

#### 4.2.3. Bosphore EX-Tract Dry Swab RNA Solution

RNA was extracted from the sample using the Bosphore EX-Tract Dry Swab RNA solution as per the manufacturer’s instructions, with the following protocol modification: 20 µL of the homogenized nasopharyngeal swab samples was added to MicroAmp 8-tube strips with 20 µL Bosphore lysis buffer solution (Anatolia Geneworks, New Ash Green Longfield, England, Cat No. CS-003) to extract the RNA. The extracted sample was placed on the Applied Biosystems thermal cycler (Thermo Fisher Scientific, Waltham, MA, USA) at 95 °C (5 min) to inactivate the virus. The Applied Biosystems real-time thermal cycler (RT-PCR) instrument (Thermo Fisher Scientific, Waltham, MA, USA) was used for the amplification and detection of SARS-CoV-2 target genes from the extracted RNA samples. 

#### 4.2.4. Sonicator Method

The Sonicator method was used to extract the RNA from the homogenized nasopharyngeal swab sample. The procedures described below were used to obtain a high-quality RNA extract using this extraction method: Eppendorf microtubes (Merck, Darmstadt, Germany) containing homogenized nasopharyngeal swab samples were placed on a dry heating block (Thermo Fisher Scientific, Waltham, MA, USA) at 65 °C for 10 min to inactivate the virus. Thereafter, the samples were processed for 15 min at 65 °C (40 kHz) with an Ultra Bath Sonicator (RS PRO, Europe) to lyse the cells and extract RNA. The Sonicator temperature was maintained by using a thermometer. The sonicated samples were centrifuged at 179× *g* for 1 min (Eppendorf, Fisher Scientific, USA). Approximately 20 µL of the extracted RNA supernatant was transferred into another empty MicroAmp 8-tube strip (Applied Biosystem, Thermo Fischer Scientific, China), and the SARS-CoV-2 target genes were amplified and detected using a real-time thermal cycler (RT-PCR) instrument (Thermo Fisher Scientific, Waltham, MA, USA).

### 4.3. Detection of SARS-CoV-2 Genes Using RT-PCR

The Thermo Fisher TaqPath™ COVID-19 Assay Kit (Thermo Fischer Scientific, Pleasanton, CA, USA, Cat No. A47813 and A47814), an internationally approved kit, Applied Biosystems real-time thermal cycler (RT-PCR) instrument (Thermo Fisher Scientific, Waltham, MA USA), and Quant-Studio Design & Analysis Software (Thermo Fisher Scientific, Waltham, MA, USA) were used to assess the RNA yield from each extraction method for the testing of SARS-CoV-2 infection. The Thermo Fisher TaqPath™ COVID-19 Assay Kit is a multiplex diagnostic solution that contains both the assays and controls needed for the RT-PCR detection of SARS-CoV-2 viral RNA. The Thermo Fisher TaqPath™ COVID-19 Assay Kit targets the *S* gene, *N* gene, and *ORF1ab* gene of SARS-CoV-2 and *MS2* (internal control). Approximately, 6.3 µL of extracted RNA was added to 2 µL of 4X Taqpath 1-step multiplex Master Mixture (mix) (Thermo Fischer Scientific, Pleasanton, CA, USA), 0.3 µL of the probe, 21.4 µL nuclease-free water, and 2 µL MS2 (added only to the housekeeping control). The 4× Taqpath master mix (7.5 µL for 30 µL) already contains probes. The total volume used per reaction was 30 µL. Conditions for the Applied Biosystems real-time thermal cycler included one cycle of 2 min at 25 °C (incubation), 10 min at 53 °C (reverse transcription), 2 min at 95 °C (activation of the Taq DNA polymerase), followed by 40 cycles of 3 s at 95 °C (denaturation) and 30 s at 60 °C (anneal/extension). The results were analyzed using Quant-Studio Design & Analysis Software (Madison, WI, USA). 

### 4.4. Comparison of the Five Commercially Available SARS-CoV-2 Real-Time PCR Assay Kits

#### 4.4.1. Selection Criteria for RT-PCR Assay Kits

The following criteria were used to select the commercially available SARS-CoV-2 RT-PCR kits used in this study: (a) the assay kits could use RNA samples extracted using any manual nucleic acid extraction methods; (b) the assay kits could be performed on a Applied Biosystems real-time thermal cycler; (c) the assay kits were available on the market and could be obtained in less than 4 weeks; (d) diagnostic laboratories in LMICs should be able to afford the assay kits; and (e) the assay kits had already obtained CE-IVD certification.

#### 4.4.2. RT-PCR Laboratory Procedure

In the method comparison of commercially available RT-PCR SARS-CoV-2 assay kits, the evaluated Lucigen QuickExtract™ RNA Extraction Kit was chosen as the preferred method for RNA extraction. This choice was made after it was discovered in this study (Section 4.2.2) that this extraction was cheaper and faster. Five commercially available RT-PCR SARS-CoV-2 assay kits from different manufacturers were selected in this study for the method comparison including the Thermo Fisher TaqPath™ COVID-19 Assay Kit (Thermo Fischer Scientific, Pleasanton, CA, USA), an internationally approved kit, and the four alternative RT-PCR SARS-CoV-2 assay kits: Nucleic Acid COVID-19 Test Kit (SARS-CoV-2) (Wuhan Easy-diagnosis Biomedicine, Wuhan, China), abTES^TM^ COVID-19 qPCR I Kit (Anatech Instrument (PTY) LTD, Meadowbrook, Business Estate, Sloane Park, Gauteng, South Africa), PCL COVID19 Speedy RT-PCR Kit (PCL Inc. Multiplex In Vitro Diagnostic Global Leader, Seoul, South Korea), and the PCLMD nCoV One-Step RT-PCR Kit (PCL Inc. Multiplex In Vitro Diagnostic Global Leader, Seoul, South Korea). Positive and negative controls were included in each test run to ensure that the results were accurate and reliable. All of the commercial RT-PCR SARS-CoV-2 assay kits were compatible with the Applied Biosystems real-time thermal cycler (RT-PCR) instrument (Thermo Fisher Scientific, Waltham, MA, USA). 

##### Thermo Fisher TaqPath™ COVID-19 Assay Kit

The Thermo Fisher TaqPath™ COVID-19 Assay Kit (Thermo Fischer Scientific, Pleasanton, CA, USA) methodology is described in full detail in Section 2.3.

##### Nucleic Acid COVID-19 Test Kit (SARS-CoV-2)

This is a reverse transcription, multiplex, one-step RT-PCR assay kit designed to detect different SARS-CoV-2 specific target genes in a single tube well. A total of 25 µL of the reaction mixture was tested, with 20 µL of Master mix and 5 µL of extracted RNA sample or SARS-CoV-2 positive control or negative control. The Applied Biosystems real-time thermal cycler was used for the amplification and detection of the SARS-CoV-2 target genes. The condition of the PCR instrument included one cycle of 15 min at 50 °C for reverse transcription, 30 s at 95 °C for pre-degeneration; 45 cycles of 3 s at 95 °C for degeneration, and 45 s at 60 °C for annealing and extension.

##### abTES^TM^ COVID-19 qPCR I Kit

This commercially available test kit is a qualitative, multiplex real-time polymerase chain reaction (qPCR) kit that allows for the simultaneous detection of two SARS-CoV-2 specific targeted genes in a single reaction. Sample reagents required for the preparation of a 20 µL reaction mixture included 10 µL of 2× RT-PCR Master mix, 1 µL of RT/ Taq enzyme mix, 2 µL of Primer/Probe mix, 2 µL of nuclease-free water, and 5 µL of RNA Template from the patient sample or negative control or the SARS-CoV-2 positive control. The SARS-CoV-2 target genes were amplified and detected using an Applied Biosystems real-time thermal cycler. The RT-PCR instruments were set up as follows: One cycle of 10 min at 59 °C for cDNA synthesis, 2 min at 95 °C for initial denaturation, and 45 cycles of 10 s at 95 °C and 30 s at 57.5 °C for amplification and extension.

##### PCL COVID-19 Speedy RT-PCR Kit

The assay kit is a one-step multiplex RT-PCR kit designed to identify two SARS-CoV-2 target genes simultaneously in a single tube. The reaction mixture volume of 20 µL contained 5 µL of Master mix, 2 µL of Primer + Probe mix, 8 µL of nuclease-free water, and 5 µL extracted RNA sample or the negative control and positive control. Using an Applied Biosystems real-time thermal cycler, the SARS-CoV-2 target genes were amplified and identified. The RT-PCR instruments were programmed as one cycle of 5 min of cDNA synthesis at 50 °C, 2 min of initial denaturation at 95 °C, and 40 cycles of 5 s at 95 °C and 30 s at 55 °C for amplification and extension.

##### PCLMD nCoV One-Step RT-PCR Kit

The assay kit is a one-step, qualitative RT-PCR kit for the detection of SARS-CoV-2. This assay kit requires three types of master mixture for each sample being tested (one test tube for one gene). (i) PCR tube 1: The 20 µL reaction mixture consisted of 5 µL of Master mix, 2 µL Primer + Probe Mixture 1 (confirmatory target gene for SARS-CoV-2 infection), 8 µL nuclease-free water, and 5 µL RNA sample; (ii) PCR tube 2: The 20 µL reaction mixture consisted of 5 µL of Master mix, 2 µL Primer + Probe Mixture 2 (for screening), 8 µL nuclease-free water, and 5 µL RNA sample; (iii) PCR tube 3: The 20 µL reaction mixture consisted of 5 µL of Master mix, 2 µL of IC Primer + Probe mix (internal control), 8 µL of nuclease-free water, and 5 µL RNA sample. The PCR reaction was performed using an Applied Biosystems real-time thermal cycler under the following conditions: One cycle of 30 min at 50 °C (cDNA synthesis), 10 min at 95 °C (initial denaturation) and 40 cycles of 15 s at 95 °C and 1 min at 55 °C (amplification and extension). 

### 4.5. Statistical Analysis

The data generated by the RT-PCR Quant-Studio Design & Analysis Software were analyzed using GraphPad Prism 5. Since the data had a normal distribution, the continuous variables were presented as the mean and standard deviation. The categorical variables were presented in percentages and numbers. The paired *t*-test was used to assess whether there was a statistically significant difference between the alternative RNA extraction methods and the gold standard RNA extraction method by comparing the mean Ct value of the SARS-CoV-2 target genes (*N*, *S*, *ORF1ab*) and internal control (MS2). The calculated sensitivity and specificity for each method were used to assess and compare the clinical diagnosis between the alternative methods and the internationally approved kits for testing SARS-CoV-2 infection. A *p* < 0.05 was considered as statistically significant.

## 5. Conclusions

In conclusion, the present study found that alternative methods were cheaper, simpler, and faster and that they could be used interchangeably with internationally approved kits (Thermo Fisher PureLink™ Kit and Thermo Fisher TaqPath™ COVID-19 Assay Kit). All alternative RNA extraction methods (Lucigen QuickExtract™ RNA Extraction Kit, Bosphore EX-Tract Dry Swab RNA Solution, and Sonicator method) and all four commercial RT-PCR SARS-CoV-2 assay kits (Nucleic Acid COVID-19 Test Kit (SARS-CoV-2), abTES^TM^ COVID-19 qPCR I Kit, PCL COVID-19 (Speedy RT PCR), and PCMLD nCov One-Step RT-PCR) can be recommended for routine diagnostic use because of their excellent performance in identifying positive samples. Furthermore, implementing these cost-effective alternative methods in LMIC laboratories will help to expand the testing capacity for the mass testing of SARS-CoV-2 infection. This will allow for the early detection of infected individuals in the community. As a result, controlling and preventative measures can be implemented sooner to avoid the spread of SARS-CoV-2 infection.

## Figures and Tables

**Figure 1 ijms-23-14350-f001:**
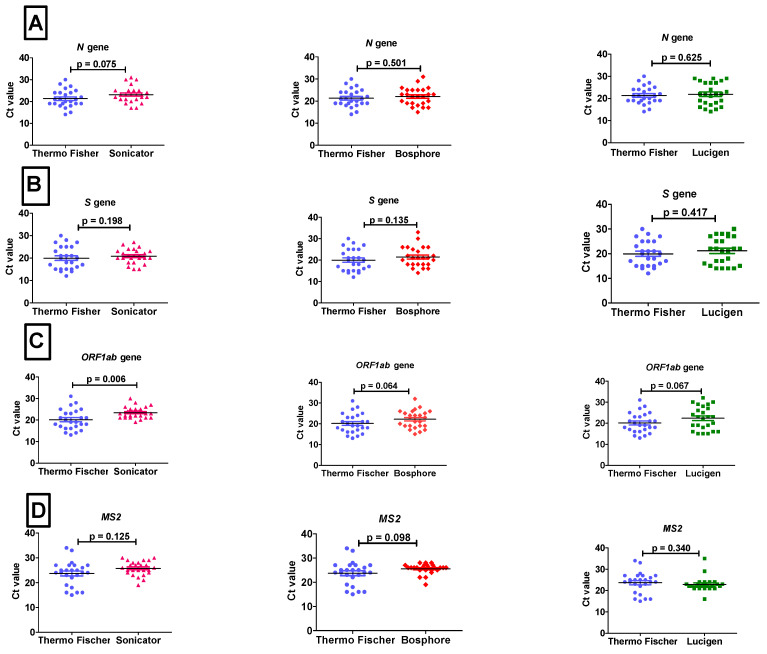
The Ct values were compared between the internationally approved kit (Thermo Fisher PureLink™ Kit) and alternative extraction methods (Sonicator method, Bosphore EX-Tract Dry Swab RNA Solution, and Lucigen QuickExtract™ RNA Extraction Kit) for the SARS-CoV-2 target genes: *N* gene (**A**), *S* gene (**B**), *Orf* gene (**C**), and MS2 internal control (**D**). The results with a level of *p* ≤ 0.05 were considered significant. Thermo Fisher PureLink^TM^ Kit is represented in blue, Sonicator method in pink, Bosphore EX-Tract Dry Swab RNA Solution in red, and Lucigen QuickExtract™ RNA Extraction Kit in green.

**Table 1 ijms-23-14350-t001:** Performance of the RNA extraction methods in testing SARS-CoV-2 infection for the positive (infected) group.

Target Genes for SARS-CoV-2 and Internal Control	Name of Extraction Method	N	Ct Mean Value (SD)	Difference between Means	*p*-Value
*N* gene	Thermo Fisher PureLink™ Kit	25	21.32 (±3.966)		
	Sonicator method	24	23.16 (±3.716)	−1.840	*p* = 0.075
	Bosphore EX-Tract Dry Swab RNA Solution	25	22.04 (±3.963)	−0.720	*p* = 0.501
	Lucigen QuickExtract™ RNA Extraction Kit	25	21.88 (±5.036)	−0.560	*p* = 0.625
*S* gene	Thermo Fisher PureLink™ Kit	25	19.96 (±5.111)		
	Sonicator method	24	20.84 (±3.197)	−0.880	*p* = 0.198
	Bosphore EX-Tract Dry Swab RNA Solution	25	21.44 (±4.647)	−1.480	*p* = 0.135
	Lucigen QuickExtract™ RNA Extraction Kit	25	21.16 (±5.421)	−1.200	*p* = 0.417
*ORF1ab* gene	Thermo Fisher PureLink™ Kit	25	20.12 (±4.702)		
	Sonicator method	24	23.40 (±2.646)	−3.280	*p* = 0.006 **
	Bosphore EX-Tract Dry Swab RNA Solution	25	22.16 (±4.249)	−2.040	*p* = 0.064
	Lucigen QuickExtract™ RNA Extraction Kit	25	22.36 (±5.469)	−2.240	*p* = 0.067
*MS2* (Internal control)	Thermo Fisher PureLink™ Kit	25	23.72 (±4.912)		
	Sonicator method	24	25.76 (±2.818)	−2.040	*p* = 0.125
	Bosphore EX-Tract Dry Swab RNA Solution	25	25.56 (±2.043)	−1.840	*p* = 0.098
	Lucigen QuickExtract™ RNA Extraction Kit	25	22.92 (±3.290)	+0.800	*p* = 0.340

Difference between means = mean Ct value of the Thermo Fisher PureLink™ Kit—mean Ct value of each alternative extraction method. ** It showed a statistically significant difference in method comparison.

**Table 2 ijms-23-14350-t002:** The sensitivity and specificity of alternative RNA extraction methods.

Name of Extraction Method	Sensitivity (*n* = 25)	Specificity (*n* = 25)
Sonicator method	24 (96%)	25 (100%)
Bosphore EX-Tract Dry Swab RNA Solution	25 (100%)	25 (100%)
Lucigen QuickExtract™ RNA Extraction Kit	25 (100%)	25 (100%)

**Table 3 ijms-23-14350-t003:** Overview of the commercially available RT-PCR assay kits used to detect the SARS-CoV-2 target genes.

SARS-CoV-2 Assay Kit	Catalog Number	Target Genes for Detection of SARS-CoV-2	Internal Control	Results Interpretation
Thermo Fisher TaqPath™ COVID-19 Assay Kit	A51738	*ORF1ab* *N* gene *S* gene	*MS2*	Ct ≤ 40 Positive result
Nucleic Acid COVID-19 Test Kit (SARS-CoV-2)	1006524-T	*N* gene *ORF1ab*	*RNase P*	Ct ≤ 40 Positive result
abTES^TM^ COVID-19 qPCR I Kit	BN300142	*NS1* *NS2*	*GAPDH*	Ct ≤ 40 Positive result
PCL COVID-19 Speedy RT-PCR kit	MD02	*N* gene *E* gene	*RNase P*	Ct < 35 Positive result
PCLMD nCoV One-Step RT-PCR Kit	MD01E	*N* gene	IC	Ct < 35 Positive result

**Table 4 ijms-23-14350-t004:** Comparison of the sensitivity and specificity of the commercial SARS-CoV-2 RT-PCR assay kits.

Name of the Kits	Number of Positive Samples Detected (*n* = 25)	False Negative	Sensitivity (%)	Number of negative Samples Detected (*n* = 25)	False Positive	Specificity (%)
Nucleic Acid COVID-19 Test Kit (SARS-CoV-2)	22	3	88	25	0	100
abTES^TM^ COVID-19 qPCR I Kit	19	6	76	24	1	96
PCL COVID-19 Speedy RT-PCR Kit	23	2	92	25	0	100
PCLMD nCoV One-Step RT-PCR Kit	24	1	96	25	0	100

**Table 5 ijms-23-14350-t005:** Comparison of the RNA extraction methods based on their simplicity cost and the observed run time for each method.

Name of Extraction Method	Simplicity	Approximate (~) Observe Run Time	* Cost per Sample
Thermo Fisher PureLink™ Kit	5 steps: Vortexing Sample lysis (Proteinase K buffer) Binding beads Washing (1st, 2nd, 3rd) Elution	~1 h	USD ~2.96
Sonicator method	4 steps: Vortexing Heating at 65 °C Sonicate at 65 °C Centrifuge	~30 min	USD ~0.18
Bosphore EX-Tract Dry Swab RNA Solution	3 steps: Vortexing Adding sample to buffer Heating sample at 95 °C	~15 min	USD ~0.89
Lucigen QuickExtract™ RNA Extraction Kit	3 steps: Vortexing Adding sample to buffer Heating sample at 95 °C	~15 min	USD ~0.59

* The cost per sample for the RNA extraction methods includes the extraction kit, reagents, consumables and excludes equipment. The ZAR to USD exchange rate: ZAR 1 = USD 0.0592 [27].

**Table 6 ijms-23-14350-t006:** Overview of the commercially available RT-PCR assay kits that were evaluated in terms of the running time, costs, and the number of SARS-CoV-2 target genes.

Name of SARS-CoV-2 Assay Kits	Running Time PCR (min)	* Cost per Sample
Thermo Fisher TaqPath™ COVID-19 Assay Kit	~64 min	USD ~14.80
Nucleic Acid COVID-19 Test Kit (SARS-CoV-2)	~83 min	USD ~4.44
abTES^TM^ COVID-19 qPCR I Kit	~79 min	USD ~9.83
PCL COVID19 Speedy RT-PCR Kit	~ 62 min	USD ~7.11
PCLMD nCoV One-Step RT-PCR Kit	~137 min	USD ~8.88

* The cost per sample includes the commercially available RT-PCR SARS-CoV-2 assay kits reagents, consumables, and sample processing and excludes equipment. The ZAR to USD exchange rate: ZAR 1 = USD 0.0592 [27]. Overall running time PCR includes the ramp time and run cycles for each RT-PCR assay kit.

## Data Availability

Not applicable.

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
