# Peer review of "Evaluation of Various Alternative Economical and High Throughput SARS-CoV-2 Testing Methods within Resource-Limited Settings"

_ijms, 2022, doi:10.3390/ijms232214350_

Round 1
Reviewer 1 Report
In the manuscript titled “Evaluation of various alternative economical and high throughput SARS-CoV-2 testing methods within resource-limited settings” by Duma et al, authors evaluate alternative methods for RT-PCR mass testing in LIMCs. Different alternative extraction methods (sonicator or two commercial lysis buffer solutions) were compared with the gold standard Thermo Fisher PureLink method. In addition, sensitivity and specificity of four commercial SARS-CoV-2 assay kits were assessed and compared with the gold standard Thermo Fisher TaqPath COVID-19 assay kit. The comparison of alternative methods can help LIMCs to the elaboration of efficient and cheaper mass testing is a really important topic. The present study highlight three good RNA extraction alternatives (cheaper and faster) and compare RT-PCR kits (efficiency, price and speed). However the small number of samples associate with several problems weakened the robustness of the study.
Major comments:
1/ In §2.3.” Thermo Fisher TaqPath™ COVID-19 assay kit was a multiplex diagnostic solution that contains both the assays and controls needed for the RT-PCR detection of SARS-CoV-2 viral RNA. Thermo Fisher TaqPath™ COVID-19 assay kit targets S gene, N gene, ORF1ab gene of SARS-CoV-2 and MS2 gene (internal control). Approximately, 6.3μLof extracted RNA was added to 2μL of 1X Taqpath 1-step multiplex Master Mixture (mix) (Thermo Fischer Scientific, Pleasanton, CA, USA), 0.3μL of the probe and 2μL MS2... When the internal control was negative, the results were considered invalid due to the inefficiency of the RNA extraction method. Figure 1 illustrates a summary of three alternative RNA extraction methods compared to the Thermo Fisher PureLink™ kit.”
Internal control MS2 of the Thermo Fisher TaqPath COVID-19 assay kit used in many steps of this study was not used correctly. The MS2 bacteriophage (not gene) control need to be added to the samples before extraction of the RNA. Here, the addition of the control during the RT-PCR step could allow PCR inhibitor detection but do not evaluate the inefficiency of the RNA extraction method leading to misinterpretation of some results.
2/ The number of samples is weak for this kind of study. It can be sufficient for the RNA extraction comparison but it is too weak to establish sensitivity and specificity of a test. In addition PCR were only performed once, it is sufficient to establish diagnosis but not for a valuable Ct value comparison.
3/The Chi square statistic test used during the extraction methods comparison is not adapted for paired data.
4/ I don’t found any information about both Lucigen lysis buffer and Bosphore lysis buffer on corresponding website. It is their manufacturer names? One problem for nasopharyngeal samples is the possible presence of inhibitors. What is the impact of these reagents in inhibitors removal?
5/ Start-up cost for SARS-CoV-2 extraction methods are confusing. This price includes extraction kit, reagents and consumables but how many samples can be processed with this start up cost? Sample price depend to the same variables but very different number can be observed by comparing both startup and sample price.
Minor comments:
- Material and Methods section
all the rpm have to be transformed in g.
in §2.2.4: Correct both Epperndorf/Eppendorf
in §2.4.2. PCLMD nCoV kit: What is the role of the PPM1, is it the positive control? Why three types of master mix?
- Results section
Figure 1 have no interest. She do not illustrates a summary of alternative RNA extraction nor highlights the comparison of alternative RT-PCR kits.
Table 1: PCLMD: the target gene for the detection of SARS is the N gene
Table 2: For MS2 gene / Lucigen lysis buffer: the difference between means is +0.8 instead of -0.8
Table 3: For MS2 gene / Lucigen lysis buffer: the difference between means is -0.04 instead of -0.4
Correct SARS instead of Sars
Table 5: Proteinase K instead of Protein K
Figure 2: legend: “S” gene in italics and Orf1ab instead of Orf
Table 7: The running time indicated in the table seems overestimated when comparing to the time of cycles in the material and methods section and with time indicated by the manufacturer when available. Which part of the PCR were taken in account?
Writing size in *
- Discussion section
“When comparing the cost of the Sonicator method (~0.18 USD per sample) to the Thermo Fisher PureLink™ kit (~2.96 USD per sample), there were more than a 60% price reduction with Sonicator methods, making the Sonicator method the cheapest method”. The price reduction between the two methods is more than 16 fold leading to a 94% price reduction.
Author Response
Dear Reviewer 1
Thank you for the constructive comments. All the revised work on the manuscript has been highlighted in green.

Reviewer 2 Report
In the manuscript presented by Duma et al. the authors assessed four extraction methods and four commercially available SARS-CoV-2 diagnostic qPCR kits to assess their feasibility for viral detection. They evaluated this in 25 COVID+ nasopharyngeal residual samples from a South African laboratory. The work is relevant as it is clear that diagnostic is still essential for the management of COVID-19 pandemic worldwide, and that although rapid antigen tests are available in developed countries, these are not universally available in low-income countries where qPCR diagnostic is still the best and more accurate diagnostic strategy. The work presented by Duma et al. is important and relevant. However, the manuscript should be improved and important details added and corrected:
Major concerns:
1. The Methods section is missing several important details. The most important: are the different extraction/qPCR kits assessed in the SAME nasopharyngeal samples? Or different cohorts of patients were used in the different experiments?
2. In the same sense, how the samples were stored; it is well documented that freezing has a clear effect on Ct values of stored samples.
3. Equally important is the composition of the transport media used for nasopharyngeal swab collection, as it may be critical for downstream analysis and different formulations are used
4. It would be desirable that the authors present more information about the cohort (age, sex, outcome, etc.).
5. Would also be desirable that the authors present more details about the reagent used. For example, Lucigen has several DNA and RNA lysis buffers and in fact, DNA purification buffers from Lucigen have been used in the past for SARS-CoV-2 diagnosis. Did the authors use the DNA or RNA extraction buffers? Presenting a catalog number would clarify and help other groups to reproduce the results.
6. Regarding the sonication method, the authors should mention the exact sonication settings as well as how do they maintain the temperature at 65 ºC.
7. There is a small difference on the “extracted” RNA volume used (6.3 uL in the case of TaqPath kit vs 5 uL in all the other kits). Can the authors justify these differences and be sure that this is not affecting the final interpretation of the results? Also, in the case of TaqPath, the total volume used per reaction should be mentioned.
8. In general, the method section is too long and over descriptive and does not in the proper tone for a method section, in fact, several parts are more adequate for the Result section, that paradoxically, the Result section is lacking connecting ideas to have a better understanding and flow of the logic of the work. Major modifications of the Material and Methods and Results section should be performed.
9. In table 2 is really concerning the SD value. Experience from many laboratories around the world has demonstrated that Ct values dramatically vary between individuals (over 10-15 Ct) as a consequence of very different viral loads that can be found at any given time in any given population. It is really surprising the small dispersion of values that the authors reports (i. e. less than 1 Ct of SD for N, S, ORF1ab and MS2 genes regarding of purification method). Do the authors selected a priory nasopharyngeal samples of very similar CT and high viral titer? If this is the case, it is critical to mention this in the Method sections, as it actually, would be more informative to test the performance of the different purification / qPCR kits on low-titer samples to really assess the performance of the reagents.
10. It is really not necessary and actually misleading presenting the same data three times (Thermo Fisher – blue dots) in figure 2. The data from “Thermo Fisher”, “Sonicator”, “Bosphore” and “Lucigen” should be presented in one chart per gene
11. in table 5 it is not clear how the “start-up” was calculated and how can be so high (eg. $2070 USD for Lucigen buffer) if as mentioned in the legend, the equipment was excluded. Also, how the cost per sample was calculated? Based on how many samples?
12. The authors conclude that the four alternative methods can be used for SARS-CoV-2 detection, but their own data suggest that the PCLMD nCoV one step RT-PCR kit is superior in terms of sensitivity (96%) vs the others (76-92%).
13. Finally, one caveat of the work is the low number of COVID+ samples used (25). The authors should be more critical and discuss this as a critical limitation of their study.
The authors repetitively use the term “gold standard” to reference a particular commercial kit (Thermo Fisher brand). But this term is not universal and what is approved as a “gold standard” in one country could be not suitable for diagnostic in other. The proper use of the term “gold standard” in SARS-CoV-2 diagnostic is in reference of the use of “RT-qPCR” (in general) and not a particular kit or brand. A more correct term e.g. “internationally approved kit” “CDC-validated” or “South Africa authorized for diagnostic kit” should be used.
Minor:
1. The quality of figure 1 is not acceptable and the description of the panel (B) is missing
2. Section 2.5 is not necessary to be described as a method.
3. Table 5 format should be corrected, and the bullets substituted
Author Response
Dear Reviewer 2
Thank you for the constructive comments. All the revised work on the manuscript has been highlighted in yellow.

Round 2
Author Response
Dear Reviewer 1,
Thank you for the construction comments. All the revised work on the manuscript has been highlighted in yellow.
